# Anthocyanin Biosynthesis Associated with Natural Variation in Autumn Leaf Coloration in *Quercus aliena* Accessions

**DOI:** 10.3390/ijms232012179

**Published:** 2022-10-12

**Authors:** Xiong Yang, Ning Yang, Qian Zhang, Ziqi Pei, Muxi Chang, Huirong Zhou, Yaoyao Ge, Qinsong Yang, Guolei Li

**Affiliations:** 1Key Laboratory of Silviculture and Conservation of the Ministry of Education, College of Forestry, Beijing Forestry University, Beijing 100083, China; 2Research Center of Deciduous Oaks, Beijing Forestry University, Beijing 100083, China

**Keywords:** autumn leaf coloration, anthocyanin biosynthesis, MYB transcription factors, natural variation

## Abstract

*Quercus aliena* is an economically important tree species and one of the dominant native oak species in China. Although its leaves typically turn yellow in autumn, we observed natural variants with red leaves. It is important to understand the mechanisms involved in leaf color variation in this species. Therefore, we compared a *Q. aliena* tree with yellow leaves and three variants with red leaves at different stages of senescence in order to determine the causes of natural variation. We found that the accumulation of anthocyanins such as cyanidin 3-*O*-glucoside and cyanidin 3-*O*-sambubiglycoside had a significant effect on leaf coloration. Gene expression analysis showed upregulation of almost all genes encoding enzymes involved in anthocyanin synthesis in the red-leaved variants during the early and main discoloration stages of senescence. These findings are consistent with the accumulation of anthocyanin in red variants. Furthermore, the variants showed significantly higher expression of transcription factors associated with anthocyanin synthesis, such as those encoded by genes *QaMYB1* and *QaMYB3*. Our findings provide new insights into the physiological and molecular mechanisms involved in autumn leaf coloration in *Q. aliena*, as well as provide genetic resources for further development and cultivation of valuable ornamental variants of this species.

## 1. Introduction

Anthocyanins, the glycosides of anthocyanidins, are water-soluble flavonoids that are produced in the cytoplasm of plants and that accumulate in vacuoles. They are one of the main pigments responsible for the coloration of flowers, fruits, and leaves [1]. Additionally, they contribute to important physiological functions such as plant defense, photoprotection, free radical scavenging, and osmotic regulation [2,3]. There are six types of anthocyanins commonly found in plants: pelargonidin 3-*O*-glucosides, cyanidin 3-*O*-glucosides, delphinidin 3-*O*-glucosides, peonidin 3-*O*-glucosides, petunidin 3-*O*-glucosides, and malvidin 3-*O*-glucosides [4]. Biosynthesis of anthocyanins is an important branch of flavonoid metabolism. Their precursor, phenylalanine, is converted into anthocyanins through a step-by-step process involving phenylalanine ammonia lyase (PAL), cinnamate-4-hydroxylase (C4H), 4-coumaryl (4CL), chalcone synthase (CHS), chalcone isomerase (CHI), flavanone 3-hydroxylase (F3H), flavanone-3′-hydroxylase (F3′H), flavanone-3′,5′-hydroxylase (F3′5′H), dihydroflavonol-4-reductase (DFR), anthocyanidin synthase (ANS), and UDP 3-*O*-glucosyltransferases (UFGT) [5,6]. Anthocyanin biosynthesis is regulated by enzyme-encoding genes in the anthocyanin biosynthetic pathway, and it can be influenced by transcription factors of the R2R3-MYB and bHLH families, which interact with WD40 protein to form an MYB–bHLH–WD40 complex that regulates anthocyanin biosynthesis [1].

Genes regulating anthocyanin biosynthesis have been studied extensively in many plants, including the model plant *Arabidopsis thaliana* [7], and other plants, such as *Populus* (poplar) [8], apple [9], grape [10], strawberry [11], tomato [12], and *Dendrobium officinale* [13]. Previous studies have also examined anthocyanin biosynthesis during the process of leaf senescence. In *Liquidambar formosana*, the gene *LfMYB113* regulates color change in autumn leaves: its protein product activates the promoter regions of *LfDFR1* and *LfDFR2* [14], while the red leaves of *Pistacia chinensis* strongly express *PcMYB113* in autumn, which is significantly associated with anthocyanin content [15]. The protein encoded by *MdbHLH3*, whose expression is induced by the protein encoded by *MdABI5*, can interact with the protein encoded by *MdMYB1* to regulate anthocyanin synthesis in apple leaves [9]. The protein encoded by *MdbHLH3* also modulates leaf senescence by regulating dehydratase-enolase-phosphatase complex 1, encoded by *MdDEP1* [16]. Transcriptome analysis has also shown that the key regulators of leaf senescence in *L. formosana*-*LfWRKY75*, *LfNAC1*, and *LfMYB113* are associated with the regulation of chlorophyll degradation and anthocyanin biosynthesis [17]. Here, this is the first study addressing autumn leaf color variation in *Q. aliena*.

*Quercus aliena* is a deciduous white oak tree species belonging to the Fagaceae family. It is native to China, Japan, and South Korea, and it is distributed at an altitude of 100–2000 m. Different parts of this economically important tree species have been used as raw materials for furniture, buildings, animal feed, starch, alcohol, and medicines [18,19,20,21]. Additionally, this species has an abundance of branches and leaves, and is therefore often used for urban and rural greening [22]. During cultivation, we found three natural variants of *Q. aliena* in Beijing, China that were significantly different in terms of their autumn leaf colors: the autumn leaves of *Q. aliena* are typically yellow, but the variants we observed had red leaves. Therefore, we aimed to examine the mechanisms involved in the natural variation of autumn leaves in *Q. aliena* in order to provide new insights into the influence of anthocyanin biosynthesis on leaf coloration.

In this study, variation in autumn leaf coloration was examined by comparing contents of total anthocyanins, total carotenoids, chlorophyll a, and chlorophyll b in the leaves of a typical *Q. aliena* plant and three natural variants. We found that anthocyanin played a major role in determining autumn leaf color variation and explored the factors contributing to anthocyanin accumulation in *Q. aliena*. Our findings may provide a basis for the development and cultivation of valuable ornamental varieties in *Q. aliena*, as well as new insights into the molecular mechanisms of anthocyanin biosynthesis during autumn leaf senescence.

## 2. Results

### 2.1. Changes of Autumn Leaves in Q. aliena Accessions

In our previous study, specific red-leaf variants were obtained through selective breeding of seedlings. As the serious introgression in *Quercus* species, molecular marker analysis was performed to explore the relationships of the four trees. The three red-leaved variants belonged to *Q. aliena*, and had a close relationship with QAC (Appendix A). When compared with the populations of *Q. aliena* and its related species, the three variants and QAC all belonged to the *Q. aliena* populations, and had divergent relationships with other species in *Quercus*, such as *Q. mongolica*, *Q. dentata*, and *Q. variabilis*. There was no significant difference in autumn leaves length, width, and area (Appendix A), and only special leaf colors were found in the four trees, which indicated that leaf color variations do not cause or accompany any other growth changes in leaf. Based on the RHS color chart, the leaf color of the red-leaved variants corresponded to the greyed-purple group, while the leaf color of the yellow-leaved tree corresponded to the greyed-orange group (Figure 1). In order to examine whether the reddening of leaves had an effect on photosynthetic capacity, we calculated Fv/Fm values for all four trees at three different stages of senescence (S1–S3), and found that the values tended to decrease as senescence progressed. Fv/Fm values for all four trees were significantly lower in the S3 stage than in the S1 stage, indicating higher potential maximum maximal quantum yield of PS II in the green stage (S1) than in the main discoloration stage (S3, Figure 2). At the S2 stage, QAC had a higher Fv/Fm value than the red-leaved variants (QA1, QA2, and QA3), but there were no significant differences in Fv/Fm among the four trees at S1 or S3. These results indicated that photosynthetic capacity is only affected by leaf color variation in the early discoloration stage.

### 2.2. Contents of Total Anthocyanins, Total Carotenoids, and Chlorophyll in Q. aliena Accessions

In order to explore the cause of reddening of leaves in *Q. aliena*, we measured the content of total anthocyanins, total carotenoids, chlorophyll a, and chlorophyll b at S1, S2, and S3 in all four trees. Compared to QAC, we observed an upward trend in the total anthocyanin content of the red-leaved variants as senescence progressed. At S3, the QA1, QA2, and QA3 leaves had significantly higher anthocyanin content than the QAC leaves, while at S1, there were no significant differences in anthocyanin content among the four accessions (Figure 3a). In fact, at S3, the anthocyanin content in QA1, QA2, and QA3 was 72–163 times higher than that measured at S1 (Figure 3a). On the other hand, we observed a downward trend in chlorophyll a and chlorophyll b content with the progression of senescence in all four trees. At S1, the content of chlorophyll a was 5–28 times higher than that measured at S3, while the content of chlorophyll b was 5–24 times higher than that measured at S3 (Figure 3b,c). In addition, at S3, the content of chlorophyll a and chlorophyll b in QAC was slightly higher than that in red-leaved variants, especially than QA1 (Figure 3b,c). In terms of total carotenoids, QA1, QA2, and QA3 leaves had slightly lower levels of carotenoids than QAC at S2 and S3 (Figure 3d). Based on the above, we believed that the leaf color variation in *Q. aliena* is mainly caused by the of anthocyanin accumulation.

Based on our analysis of individual anthocyanin components in the autumn leaves of the four *Q. aliena* accessions, we found substantially higher levels of cyanidin 3-*O*-glucoside and cyanidin 3-*O*-sambubioside in the QA1, QA2, and QA3 samples than in QAC (Table 1). This indicates that high cyanidin concentrations may contribute to the red color observed in the leaves of these natural variants. Additionally, procyanidin B1 and procyanidin B3 levels were 2–3 times higher in QA1, QA2, and QA3 than in QAC, while high content of delphinidin 3-*O*-glucoside was observed only in QA3.

### 2.3. Expression of Genes Associated with Anthocyanin Biosynthesis in Q. aliena

In order to identify genes that may regulate red coloration in the leaves of *Q. aliena* variants, we cloned and sequenced 25 *Q. aliena* genes homologous to genes previously shown in other species to be involved in anthocyanin biosynthesis (Appendix A). Additionally, semi-quantitative and real-time quantitative PCR were performed to sequence genes associated with each of the three senescence stages (S1–S3). Among the 25 genes associated with anthocyanin biosynthesis, we found that only 18 were expressed during leaf color variation in autumn. As senescence progressed, the expression of these 18 genes in QAC changed by 0–2 fold, while significant changes were found in the relative expression of anthocyanin synthesis-related genes in QA1, QA2, and QA3 (Appendix A). At S2, we observed high expression of *QaMYB1* in all three red-leaved variants, high expression of *QaDFR1* and *QaANS1* in QA1 and QA3, high expression of *QaCHS2* and *QaMYB3* in QA1, as well as high levels of *QaMYB2* and *QaF3′H* in QA3 (Appendix A). When we compared relative expression of anthocyanin synthesis-related genes between the red-leaved variants and the yellow-leaved tree, we found that all the genes except for *Qa4CL1* were expressed at significantly higher levels in QA1 than in QAC at S2 and S3 (Appendix A). However, different gene expression patterns were found in QA2 and QA3, and only *QaCHS1*, *QaCHI*, *QaF3′H*, *QaANS1*, and *QaMYB1* were expressed at significantly higher levels in all three red-leaved variants than in QAC. Based on quantitative real-time PCR, expression of *QaCHS2*, *QaDFR1*, *QaANS1*, *QaMYB1*, and *QaMYB3* in QA1 were higher at S2 and S3 (Figure 4a) than at S1, which was consistent with the semi-quantitative PCR results. Significant changes in gene expression levels in QA2 and QA3 were also verified using quantitative PCR. Once again, when we compared the red-leaved variants and the yellow-leaved control, we found higher expression of *Qa4CL2*, *QaCHS2*, *QaCHI*, *QaF3H*, *QaF3′H*, *QaDFR1*, *QaANS1*, *QaMYB1*, and *QaMYB3* in QA1 than in QAC (Figure 4b). Although the expression patterns of anthocyanin synthesis-related genes in the three variants were not exactly consistent, the increased expression of these genes is obvious as compared to QAC. In addition, the gene sequences were also compared between the three variants and QAC. However, there were no difference in amino acid sequences in all genes, such as *QaANS1*, *QaMYB1* and *QaMYB3* (Appendix A), which indicated that the leaf color variation in *Q. aliena* is not caused by the gene body mutation in anthocyanins synthesis.

Based on the genome in our laboratory (data not shown), the promoter regions of the four key genes were analyzed. Thereinto, light-responsive, auxin and ABA-responsive elements were the common elements (Figure 5a,b). The potential *MYB* and *bHLH* binding elements were also analyzed, Myb motif (TAACTG), MYB recognition site (CCGTTG) and MYB motif ([T/C]AACCA) were found in *QaCHS1*, *QaCHI*, and *QaANS1*; and MYC motif (CAT[T/G]TG) was found in *QaCHI*, *QaCHS1* and *QaF3′H*, which suggested that these genes may be regulated by the *MYB* and *bHLH* transcription factors (Figure 5c). In order to rule out the influence of leaf senescence, we performed a correlation analysis to compare anthocyanin content and gene expression (based on semi-quantitative data) in all four trees at S2 and S3 (Table 2). At S2, most of the anthocyanin synthesis-related genes significantly correlated with *QaMYB1* expression and anthocyanin content, except for *QaPAL2*, *Qa4CL1*, *Qa4CL2*, *QaCHS1*, *QaF3′H*, and *QaMYB2*. However, at S3, expression of only *QaCHS1* and *QaF3′H* significantly correlated with anthocyanin content. Therefore, we speculated that the regulation role of *QaMYB1* in anthocyanin synthesis maybe time-specific, which needs to be further studied.

### 2.4. Homology Alignment and Phylogenetic Analysis of MYB Transcription Factors

Since *QaMYB1* and *QaMYB3* expression changed significantly during senescence, we cloned and analyzed the coding DNA sequences of these two genes from the four accessions and found no significant amino acid differences among the four encoded proteins. In addition, we constructed a phylogenetic tree with other MYB transcription factors such as *DcMYB113* [23], *PtrMYB119* [8], *FaMYB10* [24] and *MdMYB9* [25], which have been studied extensively in other species. The phylogenetic analysis showed that *QaMYB1* and *QaMYB3* clustered together in the same branch with *MdMYB1*, *FaMYB10*, *AtMYB90*, and *VvMYBA1*, all of which are involved in anthocyanin biosynthesis (Figure 6a). In addition, *QaMYB2* clustered together with *MdMYB9*, *FaMYB9*, *FaMYB11*, and *VvMYBPA1*, which are involved in proanthocyanin synthesis in plants (Figure 6a). In fact, MYB genes related to anthocyanin synthesis in different species formed independent branches, indicating that the MYB gene family is conserved among all plant species. The results of the homology alignment analysis showed that the three MYB transcription factors of *Q. aliena* are relatively conserved at the N-terminus but substantially different at the C-terminus. All three members contain two MYB gene family-specific domains at the N-terminus of the sequence, the R2 and R3 domains, indicating that they all belong to the R2R3-MYB gene family. However, only the proteins encoded by *QaMYB2* and *QaMYB3* contain a unique bHLH transcription factor-binding domain, [D/E]Lx2[R/K]x3Lx6Lx3R, in the R3 domain (Figure 6b), which suggests that these two proteins may bind to bHLH transcription factors to form the MBW complex.

## 3. Discussion

### 3.1. Anthocyanin Accumulation Causes Reddening of Leaves in Q. aliena Variants

The immediate cause of color change in plants can be attributed to a change in the content of three main pigments: chlorophyll, carotenoids, and flavonoids [26]. Anthocyanins are major flavonoids that appear red, blue, or purple depending on the conditions [27]. In the present study, the contents of chlorophyll, carotenoids, and anthocyanins were all analyzed, and only significant differences were found in anthocyanins, which indicate the accumulation of anthocyanins, especially cyanidin 3-*O*-glucoside (the glycosylation product of cyanidin [28]), contributed to the autumn leaf coloration of the three natural variants of *Q. aliena*. We also found that anthocyanin accumulation and changes in leaf color were strongly associated with leaf senescence, consistent with previous work [29,30,31]. For example, a study of 89 deciduous tree species reported that 62 contained anthocyanins during leaf senescence [30], indicating that anthocyanin accumulation is a common phenomenon during senescence. During the process of leaf senescence in the red-leaved variants of *Q. aliena*, we found that the increase in anthocyanin content was accompanied by a decrease in chlorophyll content. However, there was no significant difference in chlorophyll content among QAC, QA2, and QA3, indicating that anthocyanin accumulation had no effect on the degradation of chlorophyll. During the early discoloration stage (S2), Fa/Fv values were lower in the red-leaved variants than in the yellow-leaved control, and we speculate that this may be due to the higher anthocyanin content. An increase in anthocyanin content can reduce the amount of light energy that can be absorbed and utilized by photosynthesis, thus slowing the rate of photosynthesis [32]. Additionally, a study of red maples showed that high anthocyanin content during leaf senescence can protect the plants from extreme environments such as low temperature and high solar radiation [33]. In the present study, leaf color in the *Q. aliena* accessions changed between mid-October and mid-November. During this period, we observed that the daily average temperature dropped substantially from 19.1 °C at the green stage (S1, late September) to 9.5 °C at the main discoloration stage (S3, early November; Appendix A). Previous studies have shown that low temperatures can promote anthocyanin accumulation and synthesis in plants such as *Brassica rapa*, apple, and red grapes [34,35,36]. Further research is required to understand whether and how low temperatures can cause changes in leaf coloration, and these insights will be useful for the cultivation and development of ornamental varieties of *Q. aliena*.

### 3.2. Changes in Anthocyanin Biosynthesis Alter Leaf Color in Q. aliena Variants

Anthocyanin biosynthesis is regulated by a series of enzymatic reactions, where an increase in the expression of key genes causes a corresponding increase in the expression of enzymes associated with the synthesis and accumulation of anthocyanins [37]. This accumulation of anthocyanin can alter plant color. Previous studies on natural variants have shown that high anthocyanin content was caused by the entire anthocyanin biosynthetic pathway rather than specific genes. Red variants of radish were found to express more PAL, C4H, 4CL, CHS, CHI, F3H, DFR, LDOX, and UGT enzymes than white variants, and CHS plays an important role in root coloration [38]. When four variants of radish were compared, anthocyanin content was found to be associated with levels of expression of *RsUFGT*, *RsF3H*, *RsANS*, *RsCHS3*, and *RsF3**′H1*; and *RsUFGT* was found to be the key control point involved in leaf coloration [39]. Similarly, expression levels of *F3′5′H*, *DFR*, and *UFGT* correlated significantly with anthocyanin accumulation in pepper [40]. Our results show a significant upward trend in most of the anthocyanin synthesis-related genes in the red-leaved variants during the early and main discoloration stages (S2 and S3), indicating that the accumulation of anthocyanins in oak leaves was caused by upregulation of the entire anthocyanin biosynthesis pathway, consistent with the abovementioned studies. However, we observed differential expression of anthocyanin synthesis-related genes across the three red-leaved variants, which may be caused by the different coloration patterns. Additionally, similar results were also found in natural variants of apple [41] and poplar [42]. In the present study, the expression patterns of *QaCHS1*, *QaCHI*, *QaF3′H*, and *QaANS1* were significantly different across all three variants at both S2 and S3, suggesting that these genes may be the key regulators of anthocyanin synthesis in *Q. aliena*. Since a study of *Pistacia chinensis* showed that anthocyanin biosynthesis-related genes such as *CHS* and *CHI* were regulated by MYB transcription factors during autumn leaf discoloration [43]. Thus, based on the *Q. aliena* genome in our laboratory (data not shown), promoter sequences of the key genes, *QaCHS1*, *QaCHI*, *QaF3′H*, and *QaANS1*, were analyzed. Thus, potential *MYB* binding sites were found in *QaCHS1*, *QaCHI*, and *QaANS1* (Figure 4). The Myb motif and MYB recognition site are proven to play roles in dehydration stress and abscisic acid signaling though *AtMYB2* [44,45]. Additionally, the MYB motif has dual roles in water stress and flavonoid biosynthesis [44,46]. Additionally, a bHLH motif, MYC motif was found in *QaCHI*, *QaCHS1* and *QaF3′H*, which can bind to *AtMYC* to play roles in abiotic stress [47]. Future studies should investigate whether *MYB* or *bHLH* transcription factors are involved in leaf coloration in *Q. aliena*.

### 3.3. QaMYB1 Probably Regulates Anthocyanin Biosynthesis in Q. aliena

Studies based on several different species such as grape, apple, strawberry and petunia have shown that the synthesis of plant anthocyanins is closely related to MYB transcription factors [48]. In grapes, *VvMYBAs* expression is closely related to fruit coloration: the stronger the *VvMYBAs* expression, the darker the fruit [49,50]. *MdMYB10* expression correlates with anthocyanin accumulation in apples, and expression is higher in cultivars with red fruits than in those with green fruits [51]. In the present study, we used *VvMYBA1* in grape and *MdMYB10* in apple as indices and the *Q. robur* genome sequence as a reference in order to clone and sequence the *QaMYB1*, *QaMYB2*, and *QaMYB3* genes in *Q. aliena*. *QaMYB1* was strongly expressed in the leaves of all three red-leaved variants during the early (S2) and main discoloration (S3) stages. In addition, we observed that the expression of *QaMYB1* was higher in all three variants than in the yellow-leaved control (QAC). Further phylogenetic tree analysis showed that *QaMYB1* and *QaMYB3* were in the same clade as other MYB gene family members related to anthocyanin synthesis, while *QaMYB2* was in the same clade as the MYB gene family members related to proanthocyanin synthesis. These findings suggest that *QaMYB1* may play a role in autumn leaf coloration in *Q. aliena*.

Previous studies have also shown that MYB transcription factors can affect anthocyanin accumulation by binding the promoters of anthocyanin biosynthesis-related genes directly or through the MYB-bHLH-WD40 complex [52,53]. In strawberry, *FvMYB10* binds to the promoter regions of genes encoding key enzymes such as *FvCHS2* and *FvDFR1*, and it activates their expression, promoting the accumulation of anthocyanins [54]. In *Triadica sebifera*, the protein encoded by *SsMYB1* activates anthocyanin biosynthesis by binding to the promoters of *SsDFR1* and *SsANS* [55]. In kiwi fruit, the protein encoded by *AcMYBF110* plays an important role in the regulation of anthocyanin accumulation by forming the MYB-bHLH-WD40 complex [56]. In the present study, *QaMYB1* and *QaMYB3* significantly correlated with the expression levels of other anthocyanin synthesis-related genes, including *QaPAL1*, *QaCHS2*, *QaCHI*, *QaF3H*, *QaDFR1*, *QaDFR2*, *QaDFR3*, *QaUFGT1*, and *QaANS1* at the S2 stage, indicating that transcription factors may regulate the expression of anthocyanin synthesis-related genes and subsequent anthocyanin accumulation in *Q. aliena*. However, the homology alignment identified mutations in the sequence motif within the protein encoded by *QaMYB1* that may affect binding to bHLH transcription factors. Notably, we were unable to identify specific bHLH transcription factors in *Q. aliena*. Although the homologue of *MdbHLH33* could be found in *Q. robur* genome, there was no product in *Q. aliena.* Further research is required to understand how *QaMYB1* may regulate anthocyanin synthesis in *Q. aliena*. A study of *FvMYB10* in strawberry showed that mutations in *MYB10* can cause differences in anthocyanin content [57]. Although we observed no significant differences in the sequences of proteins encoded by *QaMYB1* between the yellow-leaved control and the red-leaved variants, we cannot exclude the possibility that differences in gene expression may contribute to leaf color variation. Additionally, and notably, the promoter regions of *QaMYB1* in the four trees were analyzed, and an important base mutation was found in the cis-regulatory element, aaaAaaC(G/C)GTTA (MYB binding motif involved in flavonoid biosynthetic genes regulation), which may influence the expression of *QaMYB1* (Appendix A). This possibility should be studied further. In addition, glutathione S-transferases (GSTs), anthocyanin transporters, have been shown to play important roles in anthocyanin accumulation [58]. It can also interact with the *MYB* transcription factor, such as *PpMYB10.1* [59] and *AcMYBF110* [60], to regulate the accumulation of anthocyanins. The loss of GSTs’ function can also cause the flower and fruit color variation in peach [61], or fruit color variation in apple [62]. Therefore, further exploring the mechanism of anthocyanin transport in *Q. aliena* variants has the potential roles to explain leaf color variation in autumn.

## 4. Materials and Methods

### 4.1. Plant Material

We collected plant data from four 5-year-old *Q. aliena* trees grown in the Beijing Dadongliu Nursery in Beijing, China. Three of the trees (QA1, QA2 and QA3) had red leaves and were natural variants; the fourth tree (QAC) had typical yellow leaves and was considered the control (Figure 1). All the four trees were planted together under the same conditions. Senescence in plants is typically stratified into three stages based on the degree of change in leaf color: the green stage (28 September 2020; S1), early discoloration stage (21 October 2020; S2), and main discoloration stage (27 October 2020; S3) (Figure 1). Matured leaves (fourth to sixth from the top of the branch) were collected at each of these stages from all four trees, frozen in liquid nitrogen, and stored at −80 °C until subsequent experiments.

### 4.2. Leaves Phenotypic Analysis

The color of mature leaves was determined by comparing them to the RHS Large Color Chart (MARK0011, Royal Horticultural Society, London, UK). The length, width and area of leaves were analyzed at the main discoloration stage by using CI-3000 Portable Leaf Area Meter (LI–COR, Lincoln, NE, USA). Fifteen leaves were selected for each tree.

### 4.3. Analysis of Simple Sequence Repeat Markers

Based on the previous research about *Quercus* [63], nine pairs of primers (Appendix A) were selected to analyze phylogenetic relationships among the four samples (QA1, QA2, QA3, and QAC), as well as their relationships to additional *Q. aliena* individuals, *Q. mongolica*, *Q. dentata*, and *Q. variabilis*. The simple sequence repeat procedure was performed as described [64], and the reaction products were sent to Beijing Ruibo Xingke Biotechnology Co., Ltd., Beijing, China for capillary electrophoresis analysis. Preliminary analysis of the electrophoresis results was performed using the GeneMarker v2.2.0 (SoftGenetics, LLC., State College, PA, USA), and clustering analysis was performed using the UPGMA clustering method in SAHN mode in NTSYS 2.10e [65].

### 4.4. Measurement of Chlorophyll Fluorescence

The chlorophyll fluorescence of *Q. aliena* leaves was measured using the MINI-PAM-II ultra-portable modulated chlorophyll fluorometer (Heinz Walz, Effeltrich, Germany). We calculated the ratio of variable fluorescence to maximal fluorescence (Fv/Fm) to compare the potential maximum maximal quantum yield of PS II of the leaves at S1, S2, and S3. Four leaves were selected to do analysis after adequate pre-darkening (>30 min) at all three stages.

### 4.5. Quantification of Total Anthocyanins, Total Carotenoids, Chlorophyll a, and Chlorophyll b

The extraction and quantification of total anthocyanin content from *Q. aliena* leaves were performed as described [66], with some modifications. After removing veins, leaf samples (0.4 g) were added to test tubes containing 2 mL of 1% (*v*/*v*) hydrochloric acid-methanol solution. After mixing, the samples were placed in the dark at 4 °C for 24 h to allow further extraction. Next, the mixture was centrifuged at 4 °C (12,000 rpm, 30 min), and the absorbance of the supernatant was measured at 530 nm and 657 nm using a Cary 300 UV-Vis spectrophotometer (Agilent Technologies, Santa Clara, CA, USA). Total anthocyanin content (Q_ta_) was calculated as the total anthocyanin content per gram of fresh weight (FW) using the following formula Equation (1). Individual anthocyanin components in each sample were identified and analyzed by technicians at the Wuhan Medwell Biotechnology Co. Ltd., Wuhan, China using an ultra-high performance liquid chromatography system (ExionLC™ AD, https://sciex.com.cn/, accessed on 19 November 2021) equipped with an ACQUITY BEH C18 column (1.7 µm, 2.1 × 100 mm^2^) and coupled to a mass spectrometry system (QTRAP^®^ 6500+, https://sciex.com.cn/, accessed on 19 November 2021).

Chlorophyll a, chlorophyll b, and total carotenoids were extracted and quantification from *Q. aliena* leaves as described [67], with some modifications. After removing veins, leaf samples (0.2 g) were extracted using 80% (*v*/*v*) acetone for 5 min with grinding. The mixture was then filtered into a 25-mL brown volumetric flask, diluted with 80% acetone to volume, and mixed. The absorbance of the extracts was measured at 663 nm and 646 nm using the Cary 300 UV-Vis spectrophotometer. Contents of chlorophyll a (Q_ca_), chlorophyll b (Q_cb_), and total carotenoids (Q_tc_) were calculated per gram of FW using the following formulas Equations (2)–(7). Contents of chlorophyll a, chlorophyll b, and total carotenoids were calculated per gram of FW.
(1)$$Q_{ta}=(A_{530}-0.25\times A_{657})/FW$$
(2)$$C_{ca}\;\left({\mathrm{mg}\;L}^{-1}\right)=12.21\times A_{663}-2.81\times A_{646}$$
(3)$$C_{cb}\left({\mathrm{mg}\;L}^{-1}\right)=20.13\times A_{646}-5.03\times A_{663}$$
(4)$$C_{tc}\left({\mathrm{mg}\;L}^{-1}\right)=\left(1000\times A_{470}-3.27\times C_{ca}-104\times C_{cb}\right)/229$$
(5)$$Q_{ca}\;\left({\mathrm{mg}\;g}^{-1}\right)=\left(C_{ca}\times V\times N\right)/FW$$
(6)$$Q_{cb}\;\left({\mathrm{mg}\;g}^{-1}\right)=\left(C_{cb}\times V\times N\right)/FW$$
(7)$$Q_{tc}\;\left({\mathrm{mg}\;g}^{-1}\right)=\left(C_{tc}\times V\times N\right)/FW$$

Thereinto, C_ca_ is the concentration of chlorophyll a; C_cb_ is the concentration of chlorophyll b; C_tc_ is the concentration of total carotenoids; V is the extract volume and N is the dilution factor.

### 4.6. Cloning of Genes Associated with Anthocyanin Biosynthesis

Total RNA extraction was performed based on the modified CTAB method [68], and the cDNA strand was synthesized using the Reverse Transcription System I (Promega, Madison, WI, USA). Genes involved in anthocyanin biosynthesis were identified based on previous studies (Appendix A) and the genome of *Q. robur* [69], which is closely related to *Q. aliena*. Sequences beginning 2000 bp upstream of the transcription start site were used for identification of potential cis-acting elements using the PlantCARE tool (http://bioinformatics.psb.ugent.be/webtools/plantcare/html/, accessed on 8 June 2022), and the results were visualized using TBtools (v1.098721). A phylogenetic tree was constructed using Bioedit and MEGA11.0 (https://www.megasoftware.net/home, accessed on 5 March 2022). We used the Poisson model in the Neighbor-Joining algorithm, the “complete deletion” mode, and a bootstrap value of 1000 to construct the tree [70]. The homologous sequence alignment was analyzed using DNAMAN 9.0.1.116 (Lynnon Biosoft, San Ramon, CA, USA).

### 4.7. Semi-Quantitative and Quantitative Gene Expression Analyses

According to the conserved partial sequences, primer sequences for semi-quantitative PCR and real-time quantitative PCR were designed and shown in Appendix A, and the amplicons of each primer were verified by sanger sequencing. Data collected from the semi-qPCR were photographed with a gel imager (Gel Doc^TM^ XR+, Bio-Rad, Hercules, CA, USA) and analyzed using Image Lab (version 6.0.1, Bio-Rad) in order to quantify the brightness of the product bands. The *QaACTIN* gene of *Q. aliena* was used as the internal reference [71,72]. The S1 stage samples were used as control groups when comparing the difference between leaf senescence process, and the QAC samples were used as control groups when comparing the difference between the four accessions. The gene relative expression was calculated based on the luminance ratio method. Quantitative real-time PCR was performed using the StepOne Plus^TM^ real-time PCR system with Tower detection system (ABI, Worcester, MA, USA) and the ChamQ Universal SYBR qPCR Master Mix (Vazyme Biotech Co., Ltd., Nanjing, China). Here, again, the *QaACTIN* gene was used as an internal reference, and the S1 stage and QAC samples were considered as controls. Relative gene expression was calculated using the 2^−^^∆∆Ct^ method.

### 4.8. Statistical Analyses

All data were subjected to one-way analysis of variance (ANOVA) using SPSS 22.0 (IBM, Armonk, NY, USA). Relationships between genes relative expression and anthocyanin content were analyzed between different individuals at the same stage based on Pearson’s correlation coefficients. Significance levels for two-tailed tests were defined as *p* < 0.05 and *p* < 0.01.

## 5. Conclusions

The natural variants of *Q. aliena* are a good model system to understand the mechanism involved in anthocyanin biosynthesis and their subsequent effects on autumn leaf coloration (Figure 7). Our findings indicates that the accumulation of anthocyanins, especially cyanidin 3-*O*-glucoside, is the main cause of the reddening of autumn leaves in the natural variants of *Q. aliena*. Differential expression analysis of genes related to anthocyanin biosynthesis in the four accessions indicate that the leaf color variation may be regulated by several genes, including *QaCHS1*, *QaCHI*, *QaF3′H*, and *QaANS1*. Additionally, *QaMYB1* expression is significantly associated with anthocyanin content and with expression levels of most of the anthocyanin synthesis-related genes at the early discoloration stage. The results of this study not only provide insights into the molecular and physiological causes of leaf color variation in *Q. aliena*, but also a reference for studying leaf color changes during leaf senescence.

## Figures and Tables

**Figure 1 ijms-23-12179-f001:**
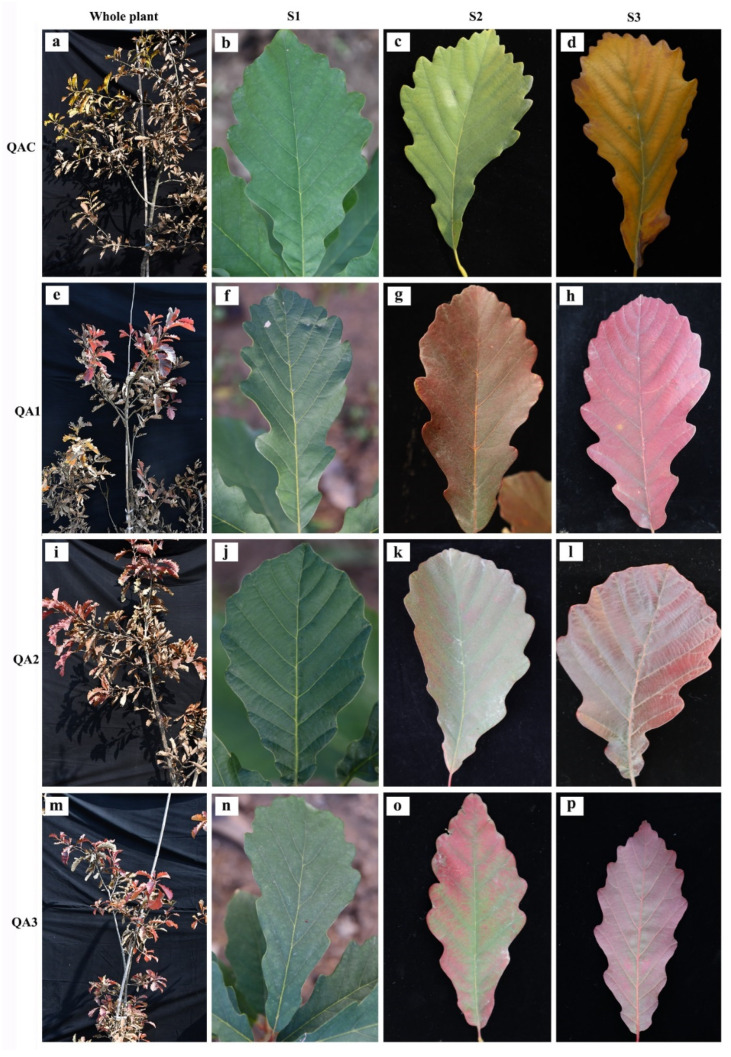
Representative images of whole plant and autumn leaf coloration in four *Quercus aliena* accessions. Whole plant images of QAC (**a**), QA1 (**e**), QA2 (**i**), and QA3 (**m**) at S3; green leaves of QAC (**b**), QA1 (**f**), QA2 (**j**), and QA3 (**n**) at S1; green-yellow leaves of QAC (**c**) at S2; green-red leaves of QA1 (**g**), QA2 (**k**), and QA3 (**o**) at S2; green-yellow leaves of QAC (**d**) at S3; and red leaves of QA1 (**h**), QA2 (**l**), and QA3 (**p**) at S3.

**Figure 2 ijms-23-12179-f002:**
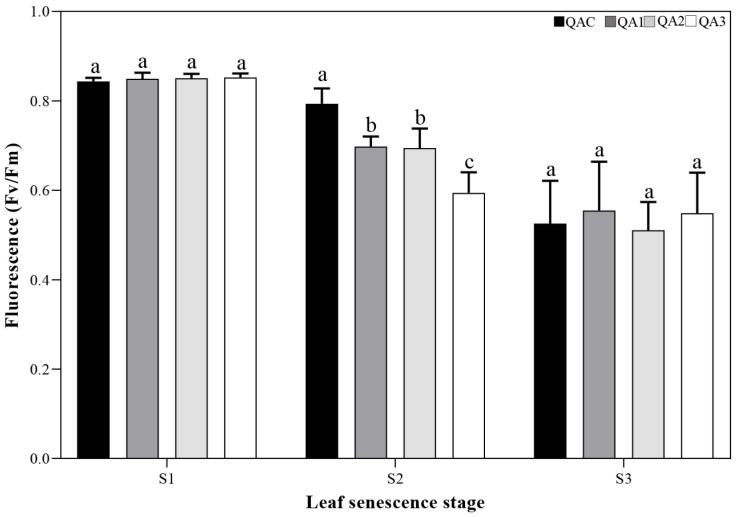
Changes in potential maximum maximal quantum yield of PS II (Fv/Fm) during leaf senescence. Bars represent the standard deviation (SD, *n* = 3), and lowercase letters represent significant differences at *p* < 0.05.

**Figure 3 ijms-23-12179-f003:**
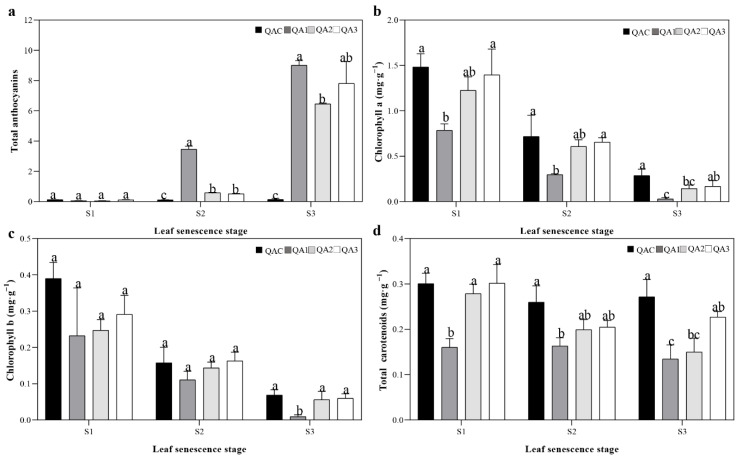
Content of (**a**) total anthocyanins, (**b**) chlorophyll a, (**c**) chlorophyll b, and (**d**) total carotenoids in *Q. aliena* accessions during leaf senescence. Bars represent standard deviation (SD), and lowercase letters represent significant differences at *p* < 0.05.

**Figure 4 ijms-23-12179-f004:**
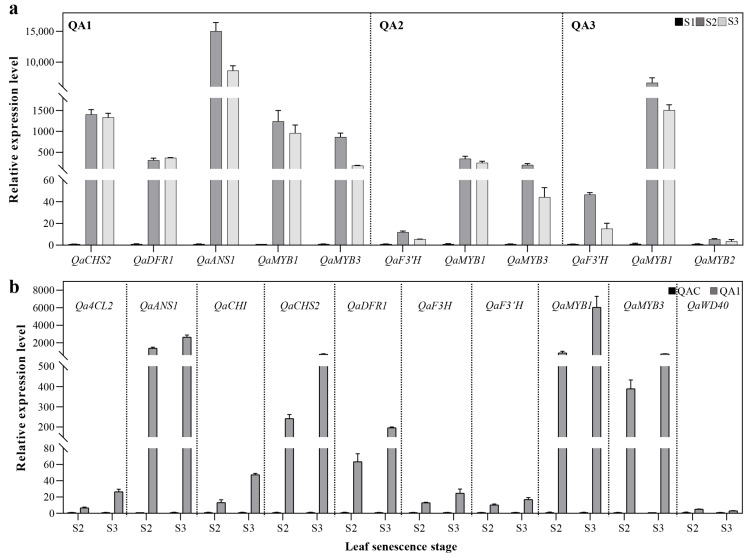
Expression of selected genes based on quantitative PCR. (**a**) Comparison of genes expressed in QA1, QA2, and QA3 during senescence (S1–S3); (**b**) comparison between QAC and QA1 at S2 and S3. Bars represent standard deviation (SD, *n* = 3).

**Figure 5 ijms-23-12179-f005:**
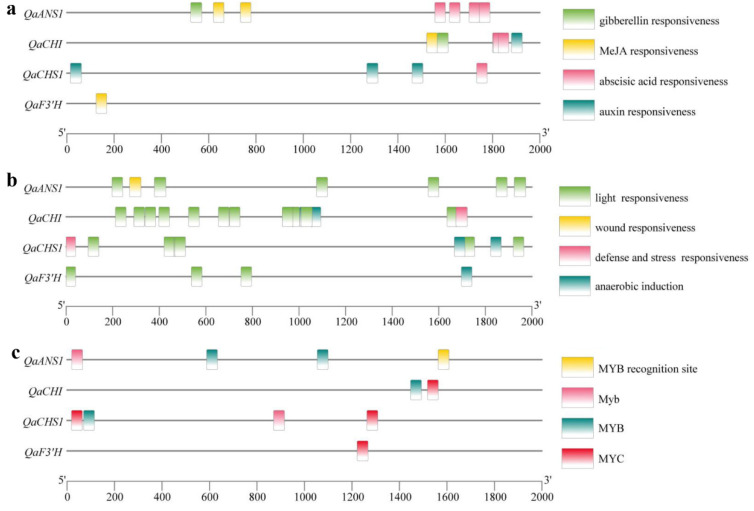
Analysis of cis-acting elements for anthocyanin synthesis-related genes in *Q. aliena*. (**a**) Plant hormone responsive elements; (**b**) environment responsive elements; (**c**) *MYB* and *bHLH* transcription factors binding elements.

**Figure 6 ijms-23-12179-f006:**
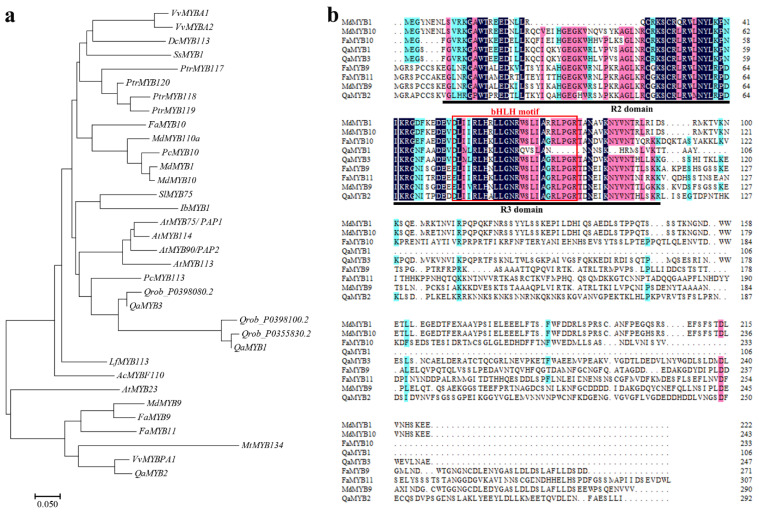
(**a**) Systematic evolution and (**b**) homology analysis of three MYB transcription factors in *Q. aliena*.

**Figure 7 ijms-23-12179-f007:**
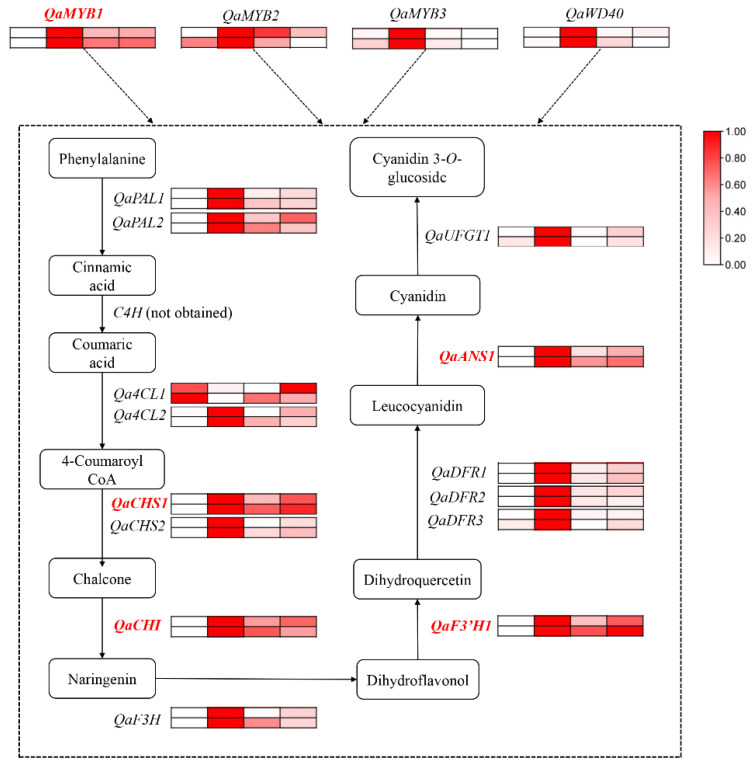
Regulatory network of anthocyanin synthesis in the leaves of *Q. aliena*. Heat maps represent the gene expression in QAC, QA1, QA2 and QA3 (left to right) at early discoloration stage (S2, the top) and main discoloration stage (S3, the bottom).

**Table 1 ijms-23-12179-t001:** Analysis of anthocyanin components in four *Q. aliena* accessions.

Compound Class	Compound	Molecular Weight (g/mol)	QAC	QA1	QA2	QA3
Cyanidin	Cyanidin 3-*O*-glucoside	449.111	1.03	777.96	662.40	583.98
Cyanidin	Cyanidin 3-*O*-sambubioside	581.15	0.03	76.49	57.03	10.56
Delphinidin	Delphinidin 3-*O*-glucoside	465.10	0.35	3.86	3.03	36.74
Procyanidin	Procyanidin B1	578.14	49.52	191.28	195.87	130.92
Procyanidin	Procyanidin B3	578.14	133.55	521.95	481.77	229.01
Flavonoid	Quercetin 3-*O*-glucoside	464.10	543.53	796.13	418.63	439.02

Values are μg/g, unless otherwise mentioned. QA1-3, red-leaved variants; QAC, typical yellow-leaved control.

**Table 2 ijms-23-12179-t002:** Correlation analysis comparing anthocyanin content and anthocyanin synthesis-related gene expression in *Q. aliena* accessions.

Gene	S2 Stage					S3 Stage				
	ACs	*QaMYB1*	*QaMYB2*	*QaMYB3*	*QaWD40*	ACs	*QaMYB1*	*QaMYB2*	*QaMYB3*	*QaWD40*
** *QaPAL1* **	0.992 **	1.000 **	0.721	1.000 **	1.000 **	0.707	0.952 *	0.865	0.980 *	0.981 *
** *QaPAL2* **	0.894	0.872	0.509	0.863	0.861	0.799	0.953 *	0.729	0.871	0.908
** *Qa4CL1* **	−0.556	−0.511	−0.959 *	−0.509	−0.511	−0.882	−0.990 **	−0.657	−0.911	−0.852
** *Qa4CL2* **	0.940	0.941	0.518	0.936	0.935	0.782	0.969 *	0.785	0.926	0.944
** *QaCHS1* **	0.880	0.852	0.516	0.841	0.839	0.998 **	0.905	0.293	0.651	0.585
** *QaCHS2* **	0.992 **	1.000 **	0.711	1.000 **	1.000 **	0.745	0.958 *	0.809	0.982 *	0.935
** *QaCHI* **	0.970 *	0.952 *	0.666	0.945	0.944	0.846	0.964 *	0.674	0.850	0.876
** *QaF3H* **	0.992 **	0.999 **	0.702	0.999 **	0.999 **	0.719	0.934	0.818	0.914	0.955 *
** *QaF3′H* **	0.946	0.935	0.569	0.929	0.927	0.961 *	0.822	0.169	0.570	0.455
** *QaDFR1* **	0.993 **	1.000 **	0.714	1.000 **	1.000 **	0.721	0.943	0.812	0.983 *	0.927
** *QaDFR2* **	0.993 **	1.000 **	0.722	1.000 **	1.000 **	0.601	0.900	0.924	0.996 **	0.991 **
** *QaDFR3* **	0.993 **	1.000 **	0.729	1.000 **	1.000 **	0.549	0.853	0.899	0.988 *	0.941
** *QaANS1* **	0.971 *	0.964 *	0.619	0.959 *	0.958 *	0.946	0.988 *	0.551	0.838	0.789
** *QaUFGT1* **	0.992 **	0.998 **	0.688	0.997 **	0.997 **	0.497	0.826	0.928	0.986 *	0.950 *
** *QaWD40* **	0.991 **	1.000 **	0.723	1.000 **		0.557	0.874	0.944	0.976 *	
** *QaMYB3* **	0.992 **	1.000 **	0.722			0.617	0.905	0.905		
** *QaMYB2* **	0.766	0.725				0.257	0.671			
** *QaMYB1* **	0.994 **					0.889				

* *p* < 0.05; ** *p* < 0.01. ACs, total anthocyanin content.

## Data Availability

The data that support the findings of this study are available in insert article and Appendix A here.

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
