# Peer review of "Anthocyanin Biosynthesis Associated with Natural Variation in Autumn Leaf Coloration in Quercus aliena Accessions"

_ijms, 2022, doi:10.3390/ijms232012179_

Round 1
Reviewer 1 Report
This manuscript is an important finding in that it identifies anthocyanin structures synthesized in red leaves of natural mutants of Quercus aliena and 25 gene sequences as genes involved in anthocyanin synthesis. These data will be useful for future variety development and mutation identification.
Unfortunately, however, the results in Fig. S1a of the evidence that QA1~3 are variants of Quercus aliena are not well explained, which confuses the reader in the introduction. QA1~3 are in the same clade as QAC, DJ, and JF, but are DJ and JF also varieties of Quercus aliena? Without a detailed explanation of which Q. mongolica, Q. dentata, and Q. variabilis they are, reader cannot understand. Likewise, what is the significance of measuring leaf size? If it is necessary for species identification, please explain in the discussion how it differs from other species.
Also, the obtained gene sequences are not registered in the database and we do not know what differences there are in the sequences between the varieties or if the sequences are identical. The sequence is shown in fasta format in Table S3, but is this the sequence obtained from QAC? From Fig. 6, QA1 is considered to have a different mutation for anthocyanin synthesis than QA2 and QA3.
Moreover, the method used to calculate the expression level amplifies the full-length sequence? If the full-length sequences were confirmed in all varieties and then the expression levels were quantified using the conserved partial sequences, this would not be a problem, but I could not find a detailed method.
The results of Figures 4 and 5 are consolidated in Figure 6 as more accurate quantitative PCR results. For this reason, Figure 4 and Figure 5 should be supplemental data. There is no need to duplicate data with the same content.
The discussion mentions bHLH, but was the bHLH sequence not obtained? Also, it is known that GSTs involved in transport are associated with anthocyanin accumulation, so it would be good to have a mention of GSTs and red leaves.
It is necessary to describe what will be considered after a careful explanation of the figures and tables. In the current manuscript, the explanations are insufficient and the contents cannot be evaluated. You should carefully check each experimental method again and reorganize the manuscript.
Reviewer 2 Report
In my opinion, the article is very interesting. Good work! I only recommend a few things:
Keywords: the same number and type of letter, please.
Lines 32 to 39: Please, re-write, remember that anthocyanins are the glycosides of anthocyanidins and the most found in plants are anthocyanins glycosides.
Line 47: “to form an MYB–47”
Line 63: “the first study”
Line 123: “chlorophyll content”
Delete the “-“ between the name of the anthocyanidin and the sugar residue, for example: “Cyanidin 3-O-glucoside”, and “-O-“ in italic
Line 148: “quantitative”
Please, meliorate the quality of the figures.
Line 226: “and only significant differences were found in anthocyanins”
Line 282: “signaling through AtMYB2”
Line 287: “regulates”
Line 338: “the top of the branch”
Line 300: “were in the same clade as other MYB gene family”
Line 344: “at the main discoloration stage”
Lines 361 and 362: “all three stages”
Line 353: “were performed”
Line 361: “pre-darkening”
Is it impossible to quantify the compounds by HPLC since this technique is more sensible?
Line 365: “leaves were performed as described”
Line 386: “total carotenoids were”
Line 411: “and real-time quantitative”
Line 434: “indicates”
Please, review all the references and put the accession numbers.

Round 2
Reviewer 1 Report
Since it is well known from other studies that bHLH is involved in anthocyanin synthesis, please describe how it was not detected in this study and your discussion.
If the O in Anthocyanidin 3-O-glucoside is in italics, then the abstract (line 21-22), introduction (line 48), conclusion (line 471), and figure 7 should also be corrected. Anthocyanidin 3-glucoside in the Introduction should also be corrected.
In Figure 7, the "n" in Dihydroquercetin has a line break, which should be corrected.
Round 3
Reviewer 1 Report
Thank you for the revision.